# Efficient Environmentally Friendly Flexible CZTSSe/ZnO Solar Cells by Optimizing ZnO Buffer Layers

**DOI:** 10.3390/ma16072869

**Published:** 2023-04-04

**Authors:** Quanzhen Sun, Jianlong Tang, Caixia Zhang, Yaling Li, Weihao Xie, Hui Deng, Qiao Zheng, Jionghua Wu, Shuying Cheng

**Affiliations:** 1College of Physics and Information Engineering, Institute of Micro-Nano Devices and Solar Cells, Fuzhou University, Fuzhou 350108, China; 2Fujian Science & Technology Innovation Laboratory for Optoelectronic Information of China, Fuzhou 350108, China; 3Jiangsu Collaborative Innovation Center of Photovoltaic Science and Engineering, Changzhou 213164, China

**Keywords:** Cd-free flexible CZTSSe/ZnO solar cells, ZnO buffer layer, CZTSSe/ZnO heterojunction, buffer layer optimization

## Abstract

Flexible CZTSSe solar cells have attracted much attention due to their earth-abundant elements, high stability, and wide application prospects. However, the environmental problems caused by the high toxicity of the Cd in the buffer layers restrict the development of flexible CZTSSe solar cells. Herein, we develop a Cd-free flexible CZTSSe/ZnO solar cell. The influences of the ZnO films on device performances are investigated. The light absorption capacity of flexible CZTSSe solar cells is enhanced due to the removal of the CdS layer. The optimal thickness of the ZnO buffer layers and the appropriate annealing temperature of the CZTSSe/ZnO are 100 nm and 200 °C. Ultimately, the optimum flexible CZTSSe/ZnO device achieves an efficiency of 5.0%, which is the highest efficiency for flexible CZTSSe/ZnO solar cells. The systematic characterizations indicate that the flexible CZTSSe/ZnO solar cells based on the optimal conditions achieved quality heterojunction, low defect density and better charge transfer capability. This work provides a new strategy for the development of the environmentally friendly and low-cost flexible CZTSSe solar cells.

## 1. Introduction

Cu_2_ZnSn(S,Se)_4_ (CZTSSe) is one of the most promising absorber materials for thin-film solar cells due to its earth-abundant, non-toxic constituent elements and tunable bandgap [1,2,3,4,5,6]. The maximum power conversion efficiency (*PCE*) of CZTSSe solar cells can reach 13.0% [7] and 11.2% [8] on rigid substrates and flexible substrates, respectively. Flexible CZTSSe solar cells have great development potential in the field of photovoltaic building integration and indoor photovoltaic applications due to their light weight and flexibility [9,10,11,12,13,14]. However, the toxic Cd element in the CdS buffer layer of the solar cell causes a series of environmental problems, limiting the application of CZTSSe solar cells. In addition, the narrow energy bandgap (E_g_) of CdS (2.4 eV) leads to loss of photons at short wavelengths (less than 520 nm), which accounts for 24% of the entire solar spectrum [15,16]. Incomplete absorption in the visible spectrum of the CdS buffer layer is thought to be responsible for the reduced quantum efficiency of CZTSSe-based solar cells [17]. Therefore, it is necessary to replace the CdS buffer layer in CZTSSe solar cells with a Cd-free and wider E_g_ buffer layer.

The ZnO semiconductor has been widely applied in a variety of solar cells, such as perovskite solar cells [18], organic solar cells [19], Sb_2_Se_3_ solar cells [20], CIGS solar cells [21], CZTSSe solar cells, etc., due to advantages which include excellent N-type characteristics, wide E_g_ (3.3 eV), non-toxicity, and abundant reserves [22]. At present, the buffer layers most commonly used in Cd-free CZTSSe solar cells with over 10% PCE are Zn_1−x_Sn_x_O (ZTO) films obtained by doping Sn elements into ZnO films [23]. Therefore, the development of quality ZnO films is key to fabricating efficient Cd-free CZTSSe devices. Compared with other Cd-free buffer layers, such as Zn_1−x_Mg_x_O, In_2_O_3_, etc. [24,25], the use of an ZnO buffer layer with a simple preparation process and low raw material cost is expected to assist in the development of low-cost environmentally friendly CZTSSe solar cells. Htay et al. used ultrasonic spray pyrolysis (USP) to deposit ZnO films on Cu_2_ZnSnS_4_ films to fabricate Cd-free solar cells based on CZTS/ZnO heterojunctions [26]. Katagiri et al. fabricated a Cd-free CZTS device with a ZnO buffer layer using atmospheric pressure chemical vapor deposition (A-CVD) [27]. In these reports, the poor density and complex preparation process of the ZnO buffer layers were not conducive to the preparation of low-cost and efficient CZTSSe/ZnO devices. Meanwhile, high-quality interfaces are very important for the improvement of device properties [28,29]. Therefore, improvements in ZnO films and the quality of CZTSSe/ZnO interfaces are beneficial to the development of low-cost and efficient CZTSSe/ZnO solar cells. However, to date, the PCEs of CZTSSe/ZnO solar cells, which are the basic component of CZTSSe/ZTO solar cells, are only 1.3% [30]. Consequently, an investigation of efficient CZTSSe/ZnO solar cells would lay the foundation for the development of low-cost and efficient Cd-free CZTSSe devices. In addition, research on Cd-free buffer layers has mainly been focused on rigid CZTSSe solar cells with soda-lime glass (SLG) as substrates, which can only be applied in limited planar scenarios, such as roof and ground situation. The application of CZTSSe solar cells could be extended to curved scenarios such as solar cars and buildings through the application Cd-free buffer layers to flexible CZTSSe solar cells.

In this work, ZnO buffer layers are used to prepare flexible CZTSSe solar cells, where the most flexible CZTSSe/ZnO device is able to achieve a PCE of 5.0%. The effects of the ZnO film thickness and the CZTSSe/ZnO heterojunction annealing temperature on the photovoltaic properties of Cd-free flexible CZTSSe solar cells are studied in detail. The results show that a flexible CZTSSe/ZnO solar cell with high-quality heterojunction and low defect density can be obtained when using the optimal ZnO buffer layer thickness (100 nm) and the appropriate annealing temperature (200 °C).

## 2. Materials and Methods

## 2.1. Fabrication of ZnO Buffer Layers

The ZnO thin film was prepared using the sputtering method, where the RF power source was applied to the ZnO target (99.99%, ZhongNuo Advanced Material (Beijing) Technology Co., Ltd., Beijing, China). The base pressure was below 9.0 × 10^−4^ Pa. The operating gas was argon. The working gas pressure was 2 Pa. The distance between the target and the sample was 12.5 cm. The deposition rate of the ZnO film was 0.05 nm/s. The thickness of the ZnO film (50, 100 and 150 nm) was controlled by the deposition time, and the flexible CZTSSe/ZnO solar cells were correspondingly named ZnO-50 nm, ZnO-100 nm and ZnO-150 nm, respectively. The annealing temperature of the CZTSSe/ZnO heterojunction was further investigated after optimization of the ZnO film thickness. The ZnO film was annealed on a hot stage at a temperature of 150–250 °C for 30 min after deposition.

## 2.2. Fabrication of Solar Cells

Flexible CZTSSe/ZnO solar cells based on Mo foil/CZTSSe/ZnO/ITO/Ag were prepared. Ethylenediamine, ethanedithiol, and elemental powders (Cu, Zn, Sn, S, and Se) were used to prepare the precursor solution by heating and stirring at 60–70 °C. The ethylenediamine, ethanedithiol, Sn powder and Se powder were purchased from Alfa Aesar. The Cu powder was purchased from Macklin. The Zn and S powders were purchased from Aladdin. Subsequently, in an Ar-filled glovebox, CZTSSe precursor films were prepared on clean Mo foils using the spin-coating method. Then, the precursor film was placed in a graphite box for the selenization process. During selenization, nitrogen was introduced into the selenization furnace to ensure that the process took place under normal pressure. The selenization conditions were raised from room temperature to 550 °C within 1 min, held at 550 °C for 900 s, and then naturally cooled to obtain the CZTSSe film (2 μm). A detailed description of the device fabrication process can be found in our previous work [31]. The ZnO and ITO (200 nm) films were deposited by RF magnetron sputtering. The Ag electrode (500 nm) was deposited by thermal evaporation. Finally, 9 cells with an active area of 0.21 cm^2^ were realized from each sample.

## 2.3. Characterizations and Measurements

The crystallization of the thin films was achieved by a multifunctional X-ray polycrystalline diffractometer (DY1602). The SEM images were characterized by a focused ion beam scanning electron microscope (Helios G4 CX). The transmittances of the films were measured using a spectrophotometer (Agilent CARY 5000 Scan, Santa Clara, CA, USA). The external quantum efficiency (EQE) spectra were measured using a QTEST HIFINITY 5 IPCE instrument. Under a standard AM1.5 illumination (100 mW/cm^2^) from a solar simulator (SUN 2000), a Keithley 2400 source meter was used to measure the current density–voltage (J-V) curves. The space charge limited current (SCLC) characterizations were conducted using a semiconductor characterization system (Fs-Pro, Hong Kong). The photoluminescence (PL) quenching spectra were measured using a Fluorescence Spectrophotometer (F-7000). Capacitance–voltage (C-V) tests were performed using a Keithley 4200 semiconductor parametric instrument. Electrochemical impedance spectra (EIS) were obtained using an electrochemical workstation (SP-200) test setup. The transient photo-voltage (TPV) and transient photocurrent (TPC) signals were recorded using an LED light source system and a digital oscilloscope (Agilent, 1 GHz), and then calculated by fitting.

## 3. Results and Discussion

In order to solve the problem related to the high toxicity of Cd in conventional flexible CZTSSe solar cells, we designed an environmentally friendly flexible CZTSSe solar cell with a device structure consisting of Mo foil/CZTSSe/ZnO/ITO/Ag (Figure 1a). Figure 1b shows a cross-section SEM image of the flexible CZTSSe device. There is a three-layer structure (CZTSSe/ZnO/ITO) on the Mo foil. The light transmittance of the buffer layers and the light absorption of CZTSSe absorber are compared between the device with CZTSSe/ZnO and the device without CZTSSe/CdS/ZnO. The ZnO and CdS/ZnO layers are grown on soda-lime glass substrates, denoted as ZnO and CdS + ZnO, respectively. Figure 1c shows the transmittance of the ZnO and CdS + ZnO layers. It can be seen that the transmittance of the buffer layer is significantly improved in the range of 400–600 nm after the removal of the CdS layer. This is because the energy band gap (E_g_) of ZnO (3.3 eV) is wider than that of CdS (2.4 eV), which is conducive to the absorption of more photons and increasing the short-circuit current density (J_sc_) of the device. Next, the EQE curves of the flexible devices with ZnO and CdS+ZnO buffer layers are shown in Figure 1d. It can be seen that the EQE values of the Cd-free CZTSSe solar cell are obviously higher than those of the device with the CdS buffer layer in the range of 400–600 nm, which is consistent with the results obtained for buffer layer transmittance. This indicates that the flexible CZTSSe solar cell with ZnO buffer layer exhibits better light absorption capacity. The integrated *J_sc_* of the flexible device with the ZnO buffer layer is greater than that of the flexible device with the CdS + ZnO buffer layer, which is because more photons enter the CZTSSe absorber. The electrical properties of the CdS/ZnO and ZnO buffer layers are analyzed using the space charge limited current (SCLC) characteristic, where the I-V data are acquired from electron-only devices with the structure of Au/buffer layer/Au. The current and voltage of the buffer layers are in accordance with Ohm’s law (I = k*V, k is constant) at low voltage. As shown in Figure 1e, the conductivity (σ) values of ZnO and CdS/ZnO buffer layers are 1.56 × 10^−8^ and 2.55 × 10^−7^ S·cm^−1^, respectively. The lower σ of the ZnO buffer layer is not conducive to the fill factor (FF) of the device.

### 3.1. Optimization of ZnO Buffer Layer Thickness

In conventional flexible CZTSSe solar cells, the CdS and ZnO layers together constitute the N-type region of the efficient solar cells [32]. The removal of CdS film affects the device performance. In addition, after the buffer layer thickness is reduced to around 10nm, the PCE of the device is only 0.6% (Appendix A). Therefore, the thickness of the ZnO layer is increased to compensate for the buffer layer effect. ZnO films with different thicknesses (50 nm, 100 nm, and 150 nm) were deposited on the CZTSSe absorbers, where the CZTSSe/ZnO heterojunctions were annealed at low temperature (150 °C). The corresponding CZTSSe solar cells are referred to as ZnO-50 nm, ZnO-100 nm, and ZnO-150 nm, respectively. The J-V curves of the devices were determined in order to obtain photovoltaic parameters. The statistical histogram and Gaussian distribution curves of the PCEs of the three devices are shown in Figure 2a. When the thickness of ZnO films is increased from 50 nm to 150 nm, the average PCE of the devices first increases from about 2% to 4% and then decreases to about 3%. The flexible CZTSSe/ZnO solar cells with 100 nm ZnO films exhibit the best performance. The J-V curves of champion devices with different ZnO film thickness are shown in Appendix A. Both the open-circuit voltage (*V_oc_*) and *J_sc_* of the ZnO-100 nm device are higher than those of the ZnO-50 nm and ZnO-150 nm devices. The best device—the ZnO-100 nm device—achieves a PCE of 4.4%, with a V_oc_ of 287 mV, a J_sc_ of 34.9 mA/cm^2^, and an FF of 44.8%. The charge extraction efficiency of the devices with optimized ZnO layer are studied on the basis of the PL quenching spectra of CdS/ZnO and ZnO buffer layers (different thicknesses) based on CZTSSe absorbers. The CZTSSe/ZnO-100 nm device shows better PL quenching (Figure 2b), indicating that the charge extraction efficiency between CZTSSe absorber and ZnO-100 nm buffer layer is higher [33]. The PL spectra of ZnO buffer layers with different thicknesses are shown in Appendix A. The PL intensity of the ZnO-100 nm buffer layer is the lowest, reflecting that there are fewer bulk defects in the ZnO-100 nm buffer layer. In order to further study the effect of ZnO thickness on the CZTSSe/ZnO heterojunction, the C-V characteristics were determined in order to obtain information on the built-in electric field and the width of the heterojunction depletion region. Figure 2c shows the C^−2^-V curves of the CZTSSe/ZnO heterojunction solar cells, where the built-in electric field (*V*_bi_) can be obtained on the basis of the intersection of the linear region extension of the curve with the *x* axis. The V_bi_ values of the ZnO-50 nm, ZnO-100 nm and ZnO-150 nm devices are 0.30 V, 0.32 V and 0.27 V, respectively. The V_bi_ of the ZnO-100 nm device is the greatest, which is consistent with the high V_oc_ of the device. Figure 2d shows curves of the width of the depletion region (W_d_) versus carrier concentration (N_D_), which can be calculated using the C-V data [32,34]. The W_d_ of the ZnO-100 nm device reaches 0.31 μm, which is wider than those of the ZnO-50 nm and ZnO-150 nm devices (0.23 and 0.27 μm). Wider values of W_d_ are beneficial to the collection and transmission of photon-generated carriers, improving the J_sc_ of the device.

### 3.2. Optimization of Annealing Temperature of CZTSSe/ZnO Heterojunction

High-quality N-type ZnO buffer layers play an important role in the development of efficient flexible CZTSSe solar cells. The improvement of the quality of ZnO films is an effective way of enhancing the performance of flexible devices. The annealing temperature is one of the important factors affecting the growth of ZnO films [35,36]. Temperatures that are too high can destroy the structure of CZTSSe absorbers, thus deteriorating the device performance. Therefore, the annealing temperature (150 °C, 200 °C and 250 °C) of the CZTSSe/ZnO was adjusted to achieve high-quality ZnO film and a flexible device with improved performance.

AFM and SEM characterizations were performed to characterize the surface roughness and crystal morphology of the CZTSSe/ZnO layers following annealing at different temperatures. The AFM images of three ZnO films deposited on CZTSSe absorbers are shown in Figure 3a–c. The root mean square (RMS) values of the surface roughness of the CZTSSe/ZnO layers produced with three different annealing temperatures are around 60 nm, showing no obvious difference. Figure 3d–f show top-view SEM images of the ZnO films prepared on CZTSSe with annealing at different temperatures. The larger grains (about 1.5 μm diameter) are the CZTSSe absorber. The fine grains (about 10 nm) on the surface are the ZnO film. According to Figure 3d–f, the annealing temperature (150–250 °C) has little effect on the morphology of the CZTSSe absorber.

The effects of annealing temperature on the crystal structures of the CZTSSe and ZnO were analyzed by XRD characterization. Conventional XRD can only detect the phase of the CZTSSe absorber due to the ZnO film being too thin. Figure 4a shows the XRD patterns of the CZTSSe absorbers after annealing at different temperatures. The (112), (220) and (312) XRD peaks belong to the kesterite-type tetragonal phase. The characteristic peaks at 14.8° (marked ♣) and 30.1° (marked ♠) may be Sn(S,Se)_2_ and Cu(S,Se), respectively [9,37,38,39], which are secondary phases. There are no significant differences among the three samples in terms of peak position, peak intensity or half width, indicating that annealing temperature in the range 150–250 °C has not effect on the crystalline phase of the CZTSSe absorber. Subsequently, the crystal structures of the ZnO films on the CZTSSe absorber were analyzed by grazing incidence X-ray diffraction (GIXRD), as shown in Figure 4b. The two characteristic peaks perfectly match those of the ZnO crystal, among which the characteristic peak of 2θ at 34.14° is the strongest, corresponding to the (100) crystal plane of ZnO, and the characteristic peak of 2θ near 62.58° corresponds to the (103) crystal plane of ZnO [40,41]. Detailed data regarding these parameters are summarized in Appendix A. With increasing annealing temperature, the full width at half maximum (FWHM) of the characteristic peaks of XRD decreases gradually, indicating that the grain size of the ZnO films is gradually increasing. However, the characteristic peak intensities show different trends. The characteristic peak intensity is higher for the ZnO film annealed at a temperature of 200 °C compared to the films based on annealing temperatures of 150 °C and 250 °C. Therefore, the ZnO film shows good crystallinity at an annealing temperature of 200 °C.

### 3.3. Performance Characterizations of Flexible CZTSSe/ZnO Devices with Different Annealing Temperatures

The statistical parameters of flexible CZTSSe/ZnO devices produced under different annealing temperatrues are displayed in Figure 5a–d. As expected, the device parameters (V_oc_, J_sc_, FF, and PCE) show a trend of first increasing and then decreasing as the annealing temperature increases from 150 to 250 °C. The best device performance is achieved at the annealing temperature of 200 °C. Figure 5e shows the J-V curves of the optimal flexible CZTSSe/ZnO solar cells produced under different annealing temperatures, and the device parameters obtained on the basis of the J-V data are listed in Table 1. The optimal device performance is realized with the flexible CZTSSe/ZnO solar cell produced with an annealing temperature of 200 °C, the PCE of which is 5.0%, with a V_oc_ of 301.3 mV, a J_sc_ of 37.8mA/cm^2^ and an FF of 44.0%. However, annealing at an excessive temperature (250 °C) significantly reduces the device efficiency, to 1.6%. The diode parameter parameters, such as shunt conductance (G_sh_), series resistance (R_s_), diode ideality factor (A), and reverse saturation current (J_0_), can be obtained from the J-V data to analyze the reasons for the variation in device performance. It can be seen that the flexible device produced under an annealing temperatrue of 200 °C has the lowest values of G_sh_ (8.4 mS/cm^2^) and R_s_ (1.7 Ω·cm^2^), as well as a better value of A (2.4), which indicates that the quality of the CZTSSe/ZnO heterojunction is obviously improved. However, The G_sh_ of the device increases significantly from 8.4 to 60.2 mS/cm^2^ when the annealing temperature is increased to 250 °C, resulting in serious interface recombination following by a deterioriation in device performance. The EQE curves of the three flexible devices are shown in Figure 5f, with an absorption wavelength range of 350–1300 nm. Compared with the EQE values of the device produced with an annealing temperature of 150 °C, those of the device based on the annealing temperature of 200 °C demonstrate a significant improvement, and the maximum EQE value reaches about 90%. When the annealing temperature is further increased to 250 °C, the EQE of the device drops significantly, with a maximum EQE value of less than 50%. This may be because the high annealing temperature disrupts the structure of the CZTSSe/ZnO heterojunction, increasing interface recombination and thus resulting in poor performance.

The SCLC characteristics were measured using electron-only devices with a structure of Au/CZTSSe/ZnO/Au to assess the electron-trap density at the CZTSSe/ZnO interfaces modified using different annealing temperatures. The current and voltage of the CZTSSe/ZnO films are in accordance with Ohm’s law at low voltages. With increasing voltage, the defect states in the CZTSSe/ZnO film are filled. The current and voltage follow the Mott–Gurney law (I ∝ V^2^) when the voltage reaches the filling limit voltage (V_TFL_). The V_TFL_ and the defect density (N_t_) conform to Equation (1):(1)$$V_{\mathrm{TFL}}=\frac{{\mathrm{qN}}_{t}L^{2}}{2\varepsilon_{0}\varepsilon_{r}}$$
where q is the electronic charge, L is the electrode distance, ε_0_ is the vacuum dielectric constant, and ε_r_ is the relative dielectric constant. The I-V curves of CZTSSe/ZnO films produced with different annealing temperatures are shown in Figure 6a. It can be observed that the N_t_ values of CZTSSe/ZnO films produced with annealing temperatures of 150 °C, 200 °C, and 250 °C are 1.75 × 10^12^, 1.50 × 10^12^ and 1.69 × 10^12^ cm^−3^, respectively, indicating that the device produced using an annealing temperature of 200 °C demonstrates less carrier recombination. The effect of annealing treatment on the CZTSSe/ZnO heterojunction interface was analyzed on the basis of the EIS characterization [42]. The equivalent circuit model of the CZTSSe photovoltaic device is an ideal series–parallel circuit of capacitors and resistors (see inset in Figure 6b). Figure 6b shows the Nyquist plots of the devices produced with different annealing temperatures. The series resistance (R_s_) and recombination impedance (R_rec_) can be determined for high-frequency and low-frequency signals, respectively. As the annealing temperature increases from 150 °C to 200 °C, the R_rec_ value of the device increases from 1.2 kΩ to 1.4 kΩ. However, when the annealing temperature is further increased to 250 °C, the R_rec_ decreases to below 0.7 kΩ. This indicates that the highest heterojunction quality and the lowest carrier recombination is achieved by the device produced with an annealing temperature of 200 °C, whereas a higher annealing temperature may cause some defects at the CZTSSe/ZnO interface, thus increasing the carrier recombination in the device. TPV and TPC characterizations were performed to explore the effect of annealing temperature on the internal defects and carrier transportation of the device [43,44]. The charge recombination lifetime (t_r_) obtained from TPV measurement is the photo-generated carriers recombination time through internal defects in the open-circuit state. The charge transfer lifetime (t_t_) obtained from TPC measurement is the time required for photo-generated carriers to be transferred to the external circuit. As shown in Figure 6c, the t_r_ values of the flexible devices produced with annealing temperatures of 150 °C, 200 °C and 250 °C are 65, 117 and 45 μs, respectively, indicating the lowest defect density is achieved by the flexible device produced using an annealing temperature of 200 °C. The t_t_ values of the three flexible devices are 22, 15 and 18 μs, respectively (Figure 6d), indicating that the device produced with an annealing temperature of 200 °C offers better charge transfer capability.

## 4. Conclusions

In this work, we demonstrated a Cd-free flexible CZTSSe/ZnO solar cell that satisfies the aims of environmental protection and low cost. The effect of the ZnO films on the performance of the flexible device was investigated. The results show that the 100nm thick ZnO film offers the best buffer layer effect. The most appropriate annealing temperature for the ZnO for attaining high performance in the flexible devices was explored. The AFM, SEM and XRD characterizations showed that the annealing treatment (150 °C–250 °C) did not destroy the structure of the CZTSSe absorbers. Meanwhile, the ZnO buffer layer exhibited better crystallinity when using an annealing temperature of 200 °C. The optimal flexible device was able to achieve a PCE of 5%, which is the highest PCE achieved by flexible CZTSSe/ZnO solar cells. The J-V, EIS, TPV and TPC characterizations showed that the optimal flexible CZTSSe/ZnO solar cell was able to realize a high-quality heterojunction, low defect density, and improved charge transfer capability. The investigation of the ZnO buffer layer in the flexible CZTSSe/ZnO solar cells could provide new concepts in the development of low-cost, environmentally friendly and efficient CZTSSe solar cells.

## Figures and Tables

**Figure 1 materials-16-02869-f001:**
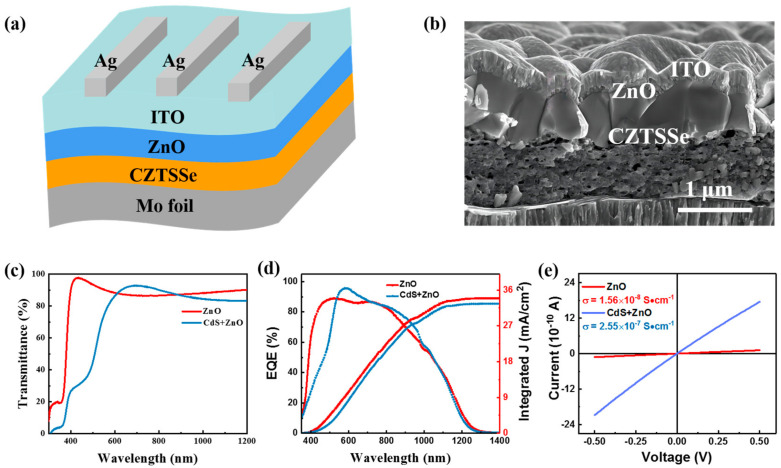
(**a**) The schematic diagram of device structure and (**b**) the cross-section SEM image for the flexible CZTSSe/ZnO solar cell. (**c**) Transmittances of the ZnO and CdS/ZnO buffer layers. (**d**) EQE curves of devices based on ZnO and CdS/ZnO buffer layers. (**e**) SCLC of ZnO and CdS/ZnO buffer layers.

**Figure 2 materials-16-02869-f002:**
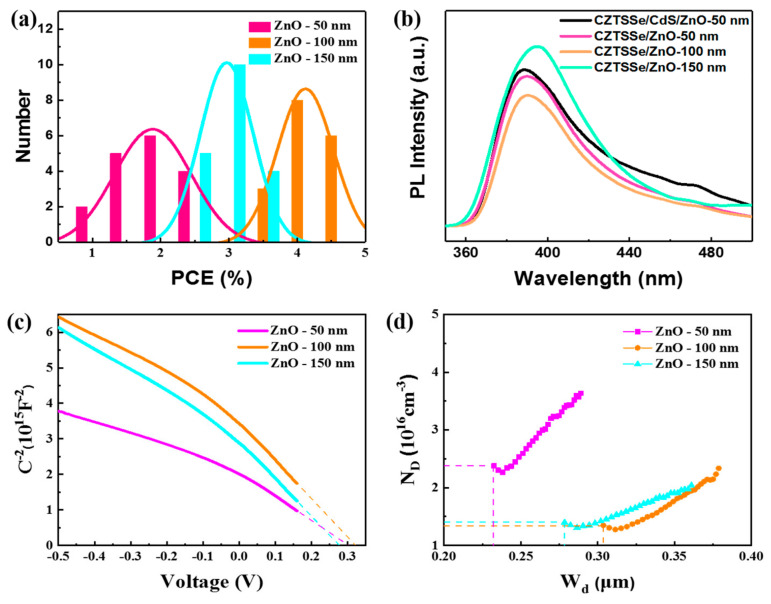
(**a**) The statistical PCE histogram of the flexible CZTSSe solar cells with ZnO layers with different thicknesses. (**b**) PL quenching spectra of the different buffer layers based on CZTSSe absorbers. (**c**,**d**) The C-V characteristics of the devices: (**c**) C^−2^-V and (**d**) N_D_-W_d_ curves.

**Figure 3 materials-16-02869-f003:**
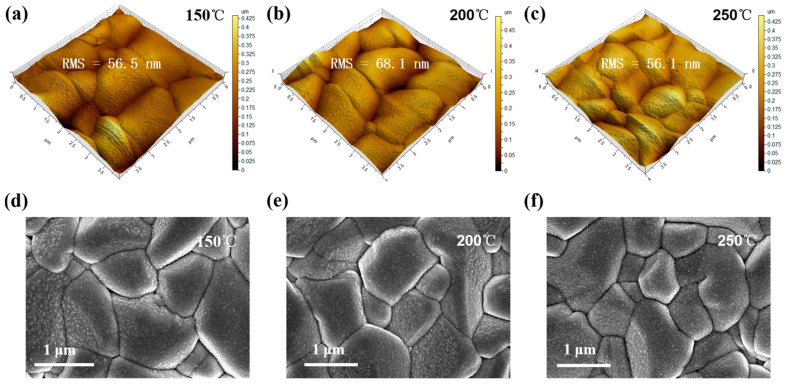
Top-view images of the ZnO films on the CZTSSe absorbers prepared at annealing temperatures of 150, 200 and 250 °C: (**a**–**c**) AFM images; (**d**–**f**) SEM images.

**Figure 4 materials-16-02869-f004:**
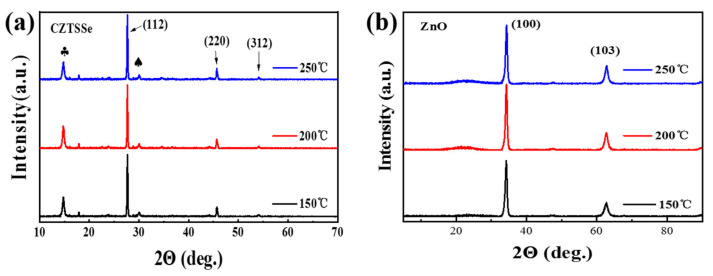
XRD patterns of ZnO/CZTSSe heterojunctions annealed at different temperatures: (**a**) conventional XRD and (**b**) GIXRD patterns. The peaks at 14.8° and 30.1° are marked ♣ and ♠, respectively.

**Figure 5 materials-16-02869-f005:**
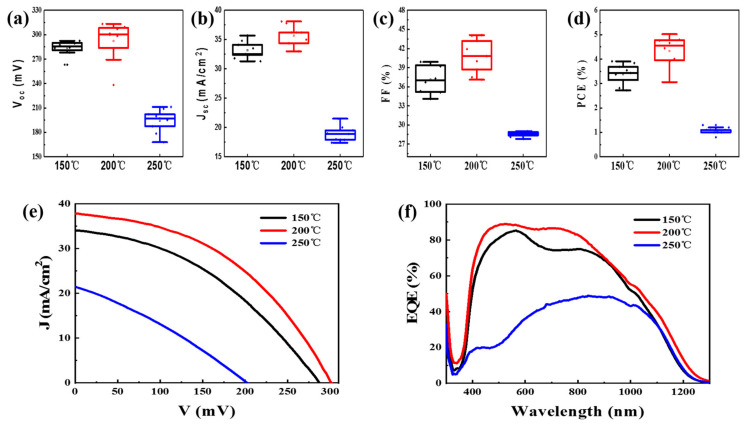
The statistical photovoltaic performance of the flexile CZTSSe solar cells based on 100 nm ZnO films produced under different annealing temperatures: (**a**) V_oc_, (**b**) J_sc_, (**c**) FF and (**d**) PCE. (**e**) The EQE and (**f**) J-V curves of the champion flexile CZTSSe/ZnO solar cells.

**Figure 6 materials-16-02869-f006:**
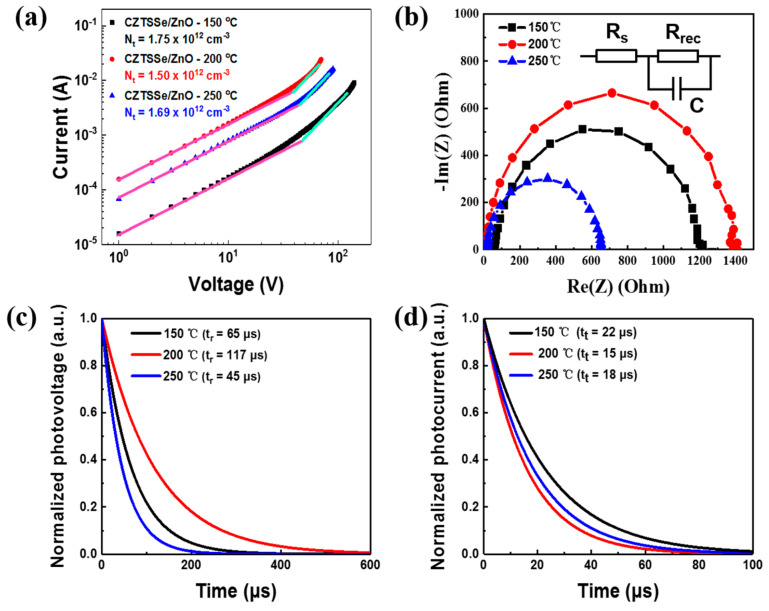
(**a**) The SCLC of CZTSSe/ZnO films produced with different annealing temperatures. The interfacial characterization of the three flexible CZTSSe/ZnO solar cells: (**b**) EIS plots, (**c**) TPV decay curves, and (**d**) TPC decay curves.

**Table 1 materials-16-02869-t001:** Performance parameters of CZTSSe/ZnO heterojunction solar cells annealed at different temperatures.

Annealing Temperature (°C)	V_oc_ (mV)	J_sc_ (mA/cm^2^)	FF (%)	PCE (%)	G_sh_ (mS/cm^2^)	R_s_ (Ω·cm^2^)	A	J_0_ (10^−3^mA/cm^2^)
150	287.0	34.9	44.8	4.4	12.0	2.2	2.6	1.8
200	301.3	37.8	44.0	5.0	8.4	1.7	2.4	2.2
250	206.1	24.3	31.7	1.6	60.2	2.8	2.6	8.4

The ZnO layer thickness in the CZTSSe/ZnO heterojunction solar cells is 100 nm.

## Data Availability

Data are contained within the article.

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
