# Peer review of "Efficient Environmentally Friendly Flexible CZTSSe/ZnO Solar Cells by Optimizing ZnO Buffer Layers"

_materials, 2023, doi:10.3390/ma16072869_

Round 1
Reviewer 1 Report
The authors developed a Cd-free flexible CZTSSe/ZnO solar cell and studied the influences of the ZnO films on the device performances. Since the CdS and ZnO layers together are generally used as buffer layer. [Solar RRL, 2020, 4.1: 1900410.] In this work the authors removed the CdS and compensate the CdS absence with thicker ZnO. There is lack of supporting data regarding to buffer layer performance and electrical properties. Authors are required to respond to following questions.
1) In Figure 1b, in the cross-section images 2 layers of porous structure and highly crystalline are identified as CZTSSe. Why does the CZTSSe active layer have 2 different textures? According to following reference the mesoporous layer is odd and need explanations. [GUO, Qijie, et al. Fabrication of 7.2% efficient CZTSSe solar cells using CZTS nanocrystals. Journal of the American Chemical Society, 2010, 132.49: 17384-17386.]
2) In this work, the CdS/ZnO was compared with the ZnO with UV-vis absorption spectra to show the preference of ZnO buffer layer over CdS/ZnO. The replacement of buffer layer cannot be only justified with one absorption spectra. The band alignment and electrical properties of buffer layer are important factors affecting the device performance. I suggest the authors compare the electrical properties of CdS/ZnO and ZnO buffer layers with SCLC analysis and electron-only devices. Also the UPS analysis can give better view about the band alignment of the buffer layers in solar cell structure.
3) Since the CdS and ZnO layers together are generally used as buffer layer. [Solar RRL, 2020, 4.1: 1900410.] The removal of CdS film was compensated with thicker ZnO layer. I suggest authors compare the PL quenching spectra of CZTSSe film with those of CdS/ZnO and ZnO buffer layers (different thicknesses) to better understand the charge extraction efficiency of optimized ZnO.
4) The effect of ZnO annealing temperature was studied. The temperature seems not to have significant effect on crystal structure of both CZTSSe and ZnO layers. However the performance fluctuate in range of 5.0%-1.6%. The authors relate the performance change to interfacial recombination. For better understanding and to assess the electron-trap density at the CZTSSe/ZnO interfaces modified by annealing temperatures, the SCLC method should be conducted for CZTSSe/ZnO films based on the electron-only devices at different temperatures.
Reviewer 2 Report
The manuscript under review presents Cd free, ZnO buffer layer based CZTSSe solar cells. The work does not highlight the novelty and similar work has already been reported earlier. Therefore, the manuscript is not recommended for acceptance.
1. There are many typo and spellings mistakes that need a thorough check.
2. The novelty in the present work is not significant and similar work has been reported previously. How is the present work, a significant addition to the already published literature about Cd free CZTSSe based solar cells?
3. In the SEM image figure 1 b, the interface between ZnO and ITO is not visible. The red and blue color is also hiding the detail.
Reviewer 3 Report
Manuscript Number materials-2266048-
Decision:
Minor Revision
Comments
The authors reported Efficient Environmental Flexible CZTSSe/ZnO Solar Cells by Optimizing ZnO Buffer Layers. The articles is very well prepared and results are very interesting. The motivation and goal of the article are well presented. This research article can be useful for the scientific community after some minor revision.
The authors should address the following points outlined below to improve the scientific quality.
1. In the introduction section author should relate the work add more important points to this study in last paragraph…. Use of interface/buffer layer is very important and highly studied topic. I suggest to add more references and compare this study with others. Cite some more references
https://www.mdpi.com/2073-4352/12/2/128
https://iopscience.iop.org/article/10.7567/JJAP.52.03BB08/meta
- In experimental section mention the company names for each chemical and equipment used in this study.
- In Materials section line 78 gas and pressure were Ar gas and 2 Pa. sentence is not clear mention the exact gases and pressure
- Line 830The ZnO film was annealed on a hot stage of 150-200 o C for 30 min after deposition. Why 150-200 temperature?
- Thickness of buffer layer was used in this experiment is 50-150 nm. Have u checked small thickness around 2-20nm?
- In section 3 results and discussion author should add Zno thichkness in table 1 also as buffer layer is main point here then the temperature.
- Line 84 author mentioned temperature range from 150-200 however in table 1 and line 209 author 250 0C .
- In AFM images add the RMS value for each temperature.
9. Last there are several grammatical errors. A proofread is required.
Round 2
Reviewer 1 Report
Authors appropriately responded the questions raised.
Reviewer 2 Report
The authors have revised the manuscript but still the novelty is not significant and similar work has been published earlier.